# Aging Process Effects on the Characteristics of Vacuum Residue Oxidation Products with the Addition of Crumb Rubber

**DOI:** 10.3390/molecules27103284

**Published:** 2022-05-20

**Authors:** Yerdos Ongarbayev, Ainur Zhambolova, Yerbol Tileuberdi, Zulkhair Mansurov, Cesare Oliviero Rossi, Pietro Calandra, Bagdat Teltayev

**Affiliations:** 1Faculty of Chemistry and Chemical Technology, Al-Farabi Kazakh National University, 71, Al-Farabi Pr., Almaty 050040, Kazakhstan; erdos.ongarbaev@kaznu.edu.kz (Y.O.); zhambolova.ainur@mail.ru (A.Z.); erbol.tileuberdi@kaznu.edu.kz (Y.T.); zmansurov@kaznu.edu.kz (Z.M.); 2Institute of Combustion Problems, 172, Bogenbaibatyr Str., Almaty 050012, Kazakhstan; 3Department of Chemistry and Chemical Technologies, University of Calabria, Via P. Bucci, Cubo 14/D, 87036 Rende, Italy; cesare.oliviero@unical.it; 4National Research Council, Institute for the Study of Nanostructured Materials (CNR-ISMN), 00015 Monterotondo, Italy; pietro.calandra@ismn.cnr.it; 5Kazakhstan Highway Research Institute, 2a, Nurpeissov Str., Almaty 050061, Kazakhstan

**Keywords:** vacuum residue, crumb rubber, bitumen, aging, oxidation, shear modulus

## Abstract

This paper considers the effect of aging processes on viscoelastic characteristics of vacuum residue oxidation products modified with crumb rubber. Viscoelastic properties were compared to original bitumen raw material-vacuum residue and vacuum residue oxidation products during short-term and long-term aging. The complex shear modulus of the vacuum residue and its oxidation products decreased with an increase in temperature. Short-term aging resulted in increased shear modulus for all samples.The vacuum residue oxidation product modified with crumb rubber had the maximum values of the rutting parameter and fatigue parameter. There was an expansion of the temperature range of plasticity: for the vacuum residue oxidation product with crumb rubber, its value was 67.2 °C. The curves of the black diagram of the modified vacuum residue oxidation product are shifted towards smaller phase angles with the increase in the shear modulus, which indicates the increase in the stiffness and elasticity of the rubber bitumen binders. The vacuum residue oxidation product modified with crumb rubber corresponded to the rubber bitumen binder of the grade RBB 60/90, according to its physical and mechanical indicators.

## 1. Introduction

Bitumen is one of the main products of the oil refining industry, but its technical characteristics do not fully satisfy the requirements of modern road construction. Special increased requirements are placed on bitumen in the construction of highways of higher technical categories and in conditions of harsh continental climates. To improve the physical and mechanical characteristics of bitumen, they are currently being modified with the addition of various fillers [1].

The study of bitumen aging is of practical importance in determining the effect of modifiers on the properties of bitumen. In the USA, a new Superpave asphalt concrete mixture design system has been developed, which considers the variation in the operational characteristics of bitumen in the process of preparing and laying asphalt concrete mixture. This asphalt laying step is already the first one in which a process of short-term aging takes place [2]. The system is then also subjected to long-term aging, with changes in its properties in the subsequent 5–10-year period of pavement operation.

The short-term ageing of bitumen occurs when mixing hot bitumen with hot mineral components during the preparation of asphalt concrete mixture and laying the pavement, which is usually carried out at a temperature of about 150 °C. To simulate aging in these processes a bitumen test is carried out using a rolling thin film oven (RTFO) [3].

Long-term aging refers to the variation in the properties of bitumen during the service life of asphalt concrete pavement. The modeling of long-term aging is carried out by the artificial oxidation of bitumen in a pressured aging vessel (PAV) [4].

When considering the effect of the structure on the properties of bitumen, particular attention is paid to its rheological properties [5]. Currently, both experimental and theoretical methods of studying of viscoelastic behavior and other rheological properties of bitumen have been developed [6]. In particular, the generalized Maxwell model is frequently used to mathematically describe the bitumen relaxation process, and it has been found that it is consistent with experimental data at temperatures from 30 to 90 °C.

In the research [7] bitumen was subjected to two oxidation processes: aging and oxygenation. It has been shown that after accelerated aging at 163 °C in RTFO and at 100 °C in PAV, oxygen absorption was observed in the samples in the form of carbonyl groups, and as the acid number increased, this absorption was due to carboxylic acids. The increase in elasticity and viscosity of the oxidation products was explained by the formation of polyaromatic compounds; polar interactions of carbonyl groups affect the phase angle and viscosity, but the main variations are made by strengthening of polyaromatic interactions.

By IR microscopy, the polymer bitumen binders were studied before and after aging to determine the polymerization rate and functional parameters characterizing bitumen, such as aromaticity, aliphaticity, and condensation [8]. Polymer modifiers are said to increase resistance to bitumen aging [9].

In the paper [10] it is established that crumb rubber as a modifier increases the stability of rubber bitumen binders during aging. The increase in crumb rubber content leads to an additional loss of the low-molecular-weight fraction of maltene from the bitumen binder during the aging process.

Rheological variations were determined to be related to bitumen aging due to the diffusion of bitumen molecules into rubber particles and the release of fillers from rubber into bitumen [11]. A comparison of the characteristic relaxation times showed that aging can’t fully explain rheological variations in the production of rubber bitumen binders.

Viscoelastic properties of the bitumen that are modified by a crumb rubber with CR30 antioxidants are determined by rheological measurements [12]. The adding of 1% CR30- and 5% CR30-modified binders increased the complex shear modulus G*. Aging had a significant effect on the rheology of bitumen by increasing the complex modulus and reducing the phase angle.

When testing bitumen modified with 20% crumb rubber, the complex module decreased with increasing temperature when compared to unmodified bitumen [13]; higher loading frequencies led to an increase in the complex modulus, but its value decreased with increasing temperature when compared to unmodified bitumen; the phase angle increased with temperature and decreased with increasing load frequency; the limit temperature of the rut resistance formation of bitumen modified with crumb rubber was 78 °C, and the limit temperature for fatigue resistance was 16 °C.

Addition of the crumb rubber also affected the physical properties of the rubberized bitumen binders, increasing their elastic recovery and reducing their penetration and ductility [14]. The rubberized bitumen binder with a higher crumb rubber content had an obvious effect on the rheological properties of the bitumen, increasing the complex shear modulus G*, the storage modulus G’, the loss modulus G’’, and reducing the phase angle.

Rheological tests [15] also showed the efficiency of rubber particles as an effective modifier of bitumen binders for the preparation of asphalt concrete pavements designed for operation in various climatic zones under extreme loads.

In the above papers, the rheological properties of bitumen are studied when modified with polymers and crumb rubber, as well as after tests with short- and long-term aging. However, there is another modification option when the modifier is added not to the prepared bitumen, but to the bitumen raw material—vacuum residue, after which the oxidation process is carried out.

The present paper studied the effect of aging processes on the rheological properties (viscoelastic characteristics) of vacuum residue oxidation products without and with the addition of crumb rubber.

## 2. Results and Discussion

When oxidizing the vacuum residue at the temperature of 260 °C for 3 h without a modifier, a product was obtained that had the needle penetration depth at 25 °C-72·0.1 mm and the softening point along the ring and ball at −48.3 °C. The physical and mechanical characteristics of the vacuum residue oxidation product with additive two and 8% by weight of crumb rubber are given in Table 1. Unlike an oxidation product without a modifier, when oxidizing a crumb-rubber-modified vacuum residue, the product is characterized by increased values of needle penetration depth at 25 °C (77·0.1 mm) and softening point (52 °C). Rubber bitumen binder is also characterized by an increased elasticity (60.0 °C) and low Fraas point (−23.0 °C), which is very important when operating asphalt concrete pavements at negative temperatures. A small variation in the softening point after heating (4.2 °C) characterizes the resistance of the rubber bitumen binder to thermal oxidative destruction. The vacuum residue oxidation product modified with crumb rubber, in terms of its physical and mechanical parameters, corresponded to the grade RBB 60/90 for a rubber bitumen binder according to the standard ST RK 2028–2010.

An important mechanical property of bitumen is the ability to resist the formation of plastic deformations. The resistance of bitumen to shear strain during multiple loads can be quantitatively expressed by the complex shear modulus G*. Figure 1 shows the graphs for the dependence of complex shear modulus G* of vacuum residue and its oxidation products on its temperature before and after aging. In the initial state, the maximum shear modulus showed the vacuum residue oxidation product with crumb rubber, compared to the vacuum residue oxidation product without a modifier (Figure 1a). Bitumen with a high G* value have a high resistance to the formation of plastic strains. As expected, when the temperature rises, the shear modulus of all samples decreases. Short-term aging resulted in an increased shear modulus for all three samples. However, at the same time, its highest values were observed for the vacuum residue itself and its oxidation product. Long-term aging dramatically increased the shear modulus to 10,000 kPa, which also decreased as the test temperature increased. Because of long-term aging, the maximum G* values at temperatures from 7 to 13 °C have the vacuum residue oxidation product modified with crumb rubber (Figure 1b), which confirms its high resistance to deformation.

Bitumen viscoelastic behavior can be described by a phase angle δ (tan δ = G”/G’); the smaller the phase angle, the greater the elasticity and the less the plasticity. Figure 2 shows variations in the phase angle δ of samples depending on the temperature and type of aging. As can be seen from the figure, when the temperature increases, the phase angle increases. In the initial state, the vacuum residue oxidation products have a phase angle higher than the vacuum residue itself. With short-term aging in the temperature range from 46 to 70 °C, the phase angle of the vacuum residue oxidation product without a modifier decreases by 4–5 °C. The vacuum residue oxidation product also has the relatively low values of the phase angle during long-term aging. As noted above, the more elastic bitumen has a low phase angle value. Research studies [16,17] found that at high temperatures, both short-term and long-term aging increase the complex shear modulus and reduce the phase angle of bitumen.

Bitumen binders’ resistance to the rut formation and fatigue cracks were determined to evaluate the possibility of using the obtained vacuum residue oxidation products as bitumen binders on roads with different traffic volume. The ability of bitumen to be resistant to rut formation was determined at a high temperature (from 46 °C and above) on the samples unaged and aged by the RTFO method. In the process of long-term aging, the stiffness of bitumen increases and the intensive formation of plastic strains stops. Therefore, the bitumen test during short-term aging was used to determine the tendency of bitumen to form plastic strains.

The rut on a pavement is formed in the first years of operation of the pavement at high temperatures, and therefore the maximum design temperature of bitumen is determined to evaluate the resistance to rut formation, since it characterizes shear resistance. The test was carried out on a dynamic shear rheometer with a loading rate of 10 rad/s (1.59 Hz). The rutting parameter G*/sin δ was calculated. The test of unaged and aged samples in RTFO to determine the parameter G*/sin δ was carried out at temperatures from 46 to 70 °C. The maximum design temperature was taken as the temperature at which the condition is met for unaged bitumen: G*/sin δ ≥ 1.0 kPa, and for bitumen aged in RTFO, the condition is met: G*/sin δ ≥ 2.2 kPa.

Fatigue cracks are formed on the surface of a road asphalt concrete pavement at medium temperatures over a long service life, in connection with which the average design temperature of bitumen is determined. This temperature test was performed on samples aged in PAV. The fatigue parameter G*⋅sin δ, characterizing bitumen’s resistance to fatigue cracking, was calculated. The average design temperature is the temperature at which the condition is met for bitumen aged in PAV: G*⋅sin δ ≥ 5000 kPa.

Figure 3 and Figure 4 show the dependencies of G*/sin δ and G*⋅sin δ parameters on temperature to determine the maximum and average design temperatures of vacuum residue and its oxidation products without and with addition of the crumb rubber for original and aged samples in RTFO and PAV. Among the original samples, a vacuum residue oxidation product with crumb rubber has the largest rut parameter, respectively, highlighting its resistance to rut formation. A comparison of the dependencies for the rut parameter for the original and aged samples showed that the magnitude of the parameter for samples subjected to the short-term aging process increased when compared to the original samples. This confirms the fact that, during the aging of bitumen, it becomes stiffer and, accordingly, less subjected to plastic strain.

After long-term aging (Figure 4), high fatigue parameter values are also characteristic of the oxidation product with the addition of crumb rubber, which proves its high resistance to fatigue cracking.

In Figure 3 and Figure 4, the maximum and average design temperatures for the tested samples given in Table 2 were determined by the limit values of the rut parameter (≥1.0 kPa for the original and ≥2.2 kPa for the aged) and the fatigue parameter (≥5000 kPa). As can be seen from this table, the vacuum residue has the lowest design temperature of 63 °C before aging. The addition of crumb rubber and subsequent short-term aging led to the increase in the design temperature from 63 to 65.2 °C.

As a result of short-term aging, the maximum design temperature increased for all samples: vacuum residue by 2.2 °C, and for oxidation products by3–3.7 °C; the largest increase occurred in the vacuum residue oxidation product with the addition of crumb rubber: an increase from 65.1 to 68.8 °C.

For samples aged in PAV, the average design temperature decreased. To evaluate the effect of the aging process on the rheological properties of bitumen, so-called “black diagrams” were also constructed, which show the dependencies of the complex shear modulus G* on the phase angle δ. Figure 5 shows the “black diagrams” of the vacuum residue and its oxidation products without the modifier and with the addition of crumb rubber.

The “black diagrams” are presented in the form of gradually falling smooth curves, the values of the phase angle of which tend to be 90°. The curves for samples aged in RTFO have the same form as for the original samples; however, the curves are shifted towards large phase angles in the vacuum residue and its oxidation product. Unlike them, the curves are shifted towards smaller phase angles in the vacuum residue oxidation product with the crumb rubber. This shift of the “black diagram” curves after aging indicates the increase in bitumen stiffness due to the increase in the complex shear modulus and the decrease in phase angle when compared to the original sample [18].

As it is seen from Figure 5b, all curves are qualitatively very similar for all samples aged in PAV. The curves of the “black diagram” are almost similar for the vacuum residue and its oxidation product with the modifier. The vacuum residue oxidation product in the “black diagram” has a curve shifted towards a smaller phase angle.

All these clues can be rationalized in terms of their physical and chemical interactions among the crumb rubber particles and the bituminous matrix. The carbon-rich nature of crumb rubber particles can be well accommodated in the organic matrix of the bitumen, reinforcing the overall bitumen supra-molecular structure hold-up by the interactions of various strengths and taking place at various length scales [19]. This effect can take place by the rubber absorption of the light components of maltene and its consequent swelling, an effect described in [20,21]. As a result of swelling, an increase in the volume of the crumb rubber particles would take place, with a reduction in their interparticle distance (structuring), which, together with the privation of lighter components from maltene phase, would cause a hindering of the dynamical processes of the bitumen molecules (stiffening). It can be argued that aphiphilic resins typical of the bitumen composition can also have an important role in further stabilizing modifiers thanks to their already ascertained role in stabilizing clusters of a wide variety of substances (from polar [22] to inorganic [23]), dispersed in apolar matrixes.

## 3. Materials and Methods

### 3.1. Materials

A vacuum residue of Omsk oil refinery, which is used at “Asphalt Concrete-1” LLP (Almaty) for the production of oxidized bitumen, was selected as a bitumen raw material. Crumb rubber produced by “Q-Recycling” LLP (Almaty) with dispersion of up to 0.6 mm was used as a vacuum residue modifier.

### 3.2. Oxidation Process

The vacuum residue oxidation process was carried out in a laboratory installation at a temperature of 260 °C. Air supply rate is 8–10 L/min. Vacuum residue oxidation time without a modifier was 3 h. The process of modifying and oxidizing vacuum residue with the addition of crumb rubber was carried out in the following way: crumb rubber was added to the vacuum residue in an amount of 2% and stirred at the temperature of 180 °C for 0.5 h, then oxidation was carried out at 260 °C for 2 h, after which the oxidation product was stirred again with 8% crumb rubber for 0.5 h.

The vacuum residue oxidation product with the addition of crumb rubber was analyzed for all the main standard physical and mechanical characteristics.

### 3.3. Aging Process

The short-term aging of samples was carried out in a special rolling thin film oven (RTFO) according to the standard [3], which models bitumen aging during the preparation, transportation, laying, and compaction of an asphalt concrete mixture. The samples were kept in the oven at the temperature of 150 °C for 75 min, where the uniform oxidation of the sample was ensured by the continuous formation of its thin layers under the influence of heat and air.

The long-term aging of samples was carried out in a special pressure aging vessel (PAV) according to the standard [4], which models bitumen aging during the operation of an asphalt concrete pavement. The samples in the vessel were at 2070 kPa and 100 °C for 20 h.

### 3.4. Mechanical Characterization

The mechanical characteristics of the samples at temperatures from 1 to 70 °C were measured by a dynamic shear rheometer, model SmartPave 102 (Anton Paar GmbH, Graz, Austria), according to the standard [24]. Samples in the form of a circular plate with a diameter of 25 mm and a thickness of 1 mm were tested for sinusoidal variable deformation with an amplitude of 12% and a frequency of 10 rad/s. Before the test, the samples were kept at a specified temperature for at least 10 min. Based on the test results, shear stress τ, shear strain γ, and phase angle δ were measured. The value of complex shear modulus G* of the bitumen is calculated by formula [25]:G* = (τ_max_ − τ_min_)/(γ_max_ − γ_min_),(1)
where τ_max_, τ_min_ are the maximum and minimum shear stresses, respectively; γ_max_, γ_min_ are the maximum and minimum shear strains, respectively.

## 4. Conclusions

Oxidation products were obtained for the unmodified vacuum residue and the vacuum residue modified with crumb rubber, and the effect of short-term and long-term aging processes on their viscoelastic characteristics were tested. Results have shown that oxidation and modification of the vacuum residue with crumb rubber lead to an increase in the complex shear modulus during long-term aging. The oxidation product is characterized by high values of the rutting parameter and fatigue parameter, which shows their higher resistance to plastic strains and fatigue cracking. The results are rationalized in terms of the crumb rubber’s capability to establish effective interactions with the bituminous matrix, helping in the holding up of the entire supramolecular network of interacting molecules. This allows for drawing the conclusion regarding the effectiveness of the modification of the vacuum residue with the crumb rubber, which shows the possibility to obtain rubber bitumen binders with improved viscoelastic characteristics.

## Figures and Tables

**Figure 1 molecules-27-03284-f001:**
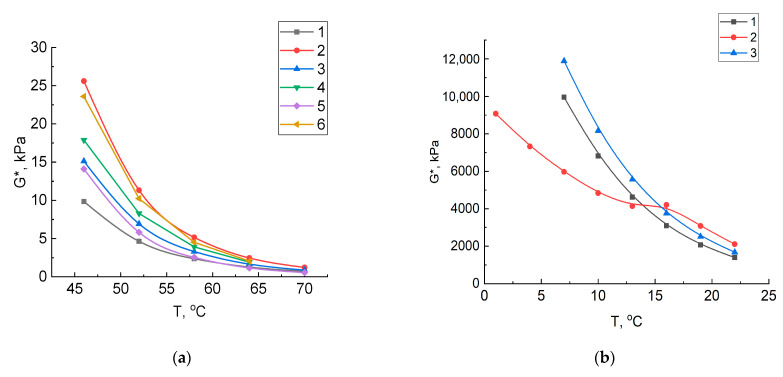
Dependences of the complex shear modulus of samples on a temperature: (**a**) vacuum residue: 1—original, 2—after aging in RTFO; vacuum residue oxidation product with crumb rubber: 3—original, 4—after aging in RTFO; vacuum residue oxidation product: 5—original, 6—after aging in RTFO; (**b**) after aging in PAV: 1—vacuum residue, 2—vacuum residue oxidation product, 3—vacuum residue oxidation product with crumb rubber.

**Figure 2 molecules-27-03284-f002:**
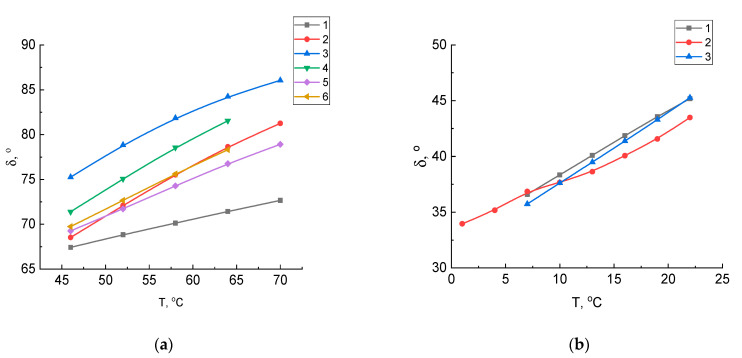
Dependences of the complex shear modulus of samples on a temperature: (**a**) vacuum residue: 1—original, 2—after aging in RTFO; vacuum residue oxidation product with crumb rubber: 3—original, 4—after aging in RTFO; vacuum residue oxidation product: 5—original, 6—after aging in RTFO; (**b**) after aging in PAV: 1—vacuum residue, 2—vacuum residue oxidation product, 3—vacuum residue oxidation product with crumb rubber.

**Figure 3 molecules-27-03284-f003:**
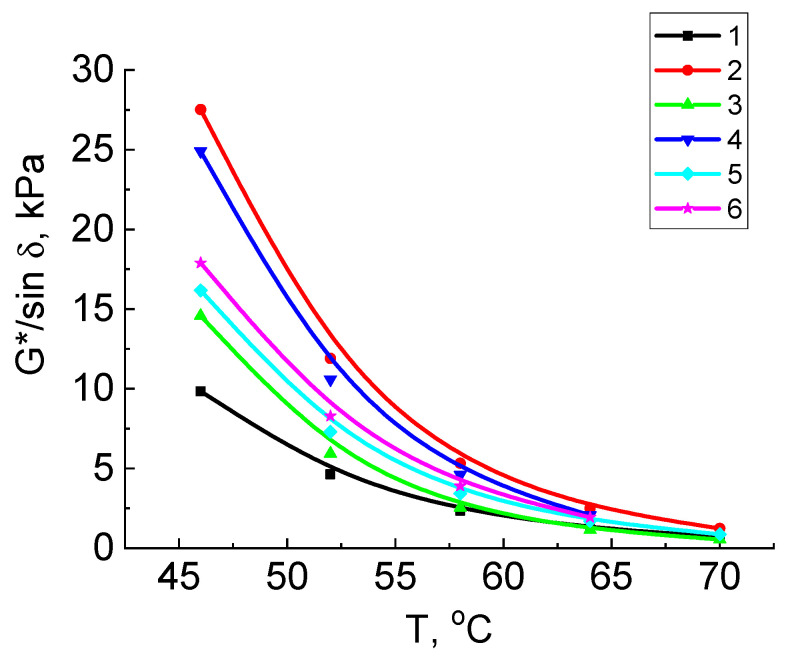
Dependences of rutting parameter G*/sin δ for samples on a temperature: vacuum residue: 1—original, 2—aged in RTFO; vacuum residue oxidation product: 3—original, 4—aged in RTFO; vacuum residue oxidation product with crumb rubber: 5—original, 6—aged in RTFO.

**Figure 4 molecules-27-03284-f004:**
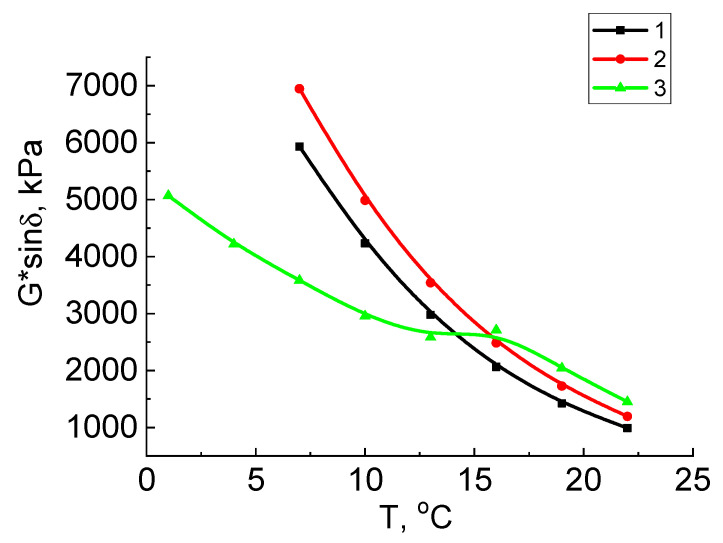
Dependences of fatigue parameter G*sin δ for samples aged in PAV on a temperature: 1—vacuum residue, 2—vacuum residue oxidation product with crumb rubber, 3—vacuum residue oxidation product.

**Figure 5 molecules-27-03284-f005:**
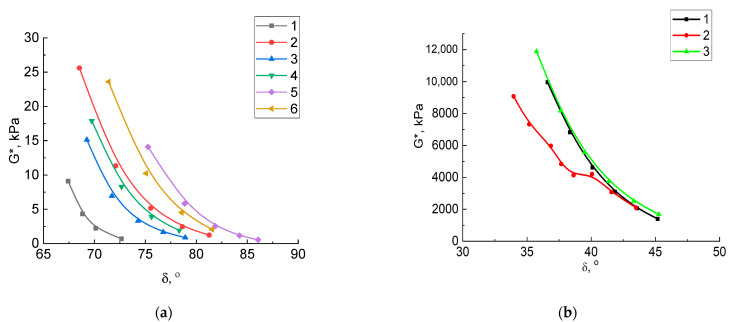
“Black diagrams”: (**a**) vacuum residue samples: 1—original, 2—aged in RTFO; vacuum residue oxidation product: 3—original, 4—aged in RTFO; vacuum residue oxidation product with crumb rubber: 5—original, 6—aged in RTFO; (**b**) after aging in PAV: 1—vacuum residue, 2—vacuum residue oxidation product, 3—vacuum residue oxidation product with crumb rubber.

**Table 1 molecules-27-03284-t001:** Physical and mechanical characteristics of rubber bitumen binder obtained by vacuum residue oxidation, modified with crumb rubber.

Indicator	Rubber Bitumen Binder	Requirements of ST RK 2028–2010 for RBB 60/90
Needle penetration depth at 25 °C, 0.1 mm	77.0	61–90
Softening point under ring and ball, °C	52.0	not below than 52
Ductility, cm: at 25 °C at 0 °C	16.0 6.0	notlessthan 12 not less than 6
Elasticity at 25 °C, °C	60.0	not less than 30
Fraas point, °C	−23.0	not higher than −18
Flashpoint, °C	260.0	not below than 250
Softening point variation after heating, °C	4.2	not more than 5

**Table 2 molecules-27-03284-t002:** Maximum and average plasticity temperatures for samples.

Sample	Maximum Design Temperature, °C		Average Design Temperature, °C
	Original	Aged in RTFO	Aged in PAV
Vacuum residue	63.0	65.2	10.0
Vacuum residue oxidation product	63.5	66.5	8.5
Vacuum residue oxidation product with crumb rubber	65.1	68.8	1.6

## Data Availability

Not Applicable.

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
