# Peer review of "Aging Process Effects on the Characteristics of Vacuum Residue Oxidation Products with the Addition of Crumb Rubber"

_molecules, 2022, doi:10.3390/molecules27103284_

Round 1

Reviewer 1 Report

The manuscript entitled Aging Process Effect on Characteristics of Vacuum Residue Oxidation Products with Addition of Crumb Rubber points out a very interesting topic related to the reuse of crumb rubber and its mixture with bitumen. However, the manuscript presents serious issues that the authors must to attend. Below are the comments.

-Authors must revise the language in the manuscript. There are several grammatical issues.

-Authors must follow the instructions for authors provided by the journal and organize the manuscript information according to it.

-Line 43. Separate the words Thesystem.

Line 67. Microscopyhe? verify the word.

-Line 75. Variationswere? idem.

-Lines 59-100. The information must be included in the Discussion section.

-Introduction section. Authors must be concise with the information provided.

-Line 113. Residueoxidation?

-Line 113. The abbreviation for liter is the capital letter L.

-Organize the materials and methods in sections. For example Oxidation process, Aging process, Mechanical characterization, etc.

-Revise table 1 information. There are grammatical mistakes in the title and in the content.

-Figures 1-5. Include standard deviation of data.

Author Response

Responses to Reviewer 1’s Comments

Comment 1: Authors must revise the language in the manuscript. There are several grammatical issues.

Response: The English language of the article has been checked and some sentences have been corrected.

Comment 2: Authors must follow the instructions for authors provided by the journal and organize the manuscript information according to it.

Response: Comment accepted. The article has been fully checked and all the manuscript information is formatted according to the requirements of the Journal’s Instructions for Authors.

Comment 3: Line 43. Separate the words Thesystem.

Response: Comment accepted. The correction has been made.

Comment 4: Line 67. Microscopyhe? verify the word.

Response: Comment accepted. The correction has been made.

Comment 5: Line 75. Variationswere? idem.

Response: Comment accepted. The correction has been made. 

Comment 6: Lines 59-100. The information must be included in the Discussion section.

Response: Here general information about the short-term and long-term aging of bitumen and about the studies of other scientists is given, indicating the literature sources. Since this information is not the result of the authors of the article, they consider it appropriate to present it in this section.

Comment 7: Introduction section. Authors must be concise with the information provided.

Response: The “Introduction” section provides some information on the topic of the article with reference to literary sources. All proposals relate to the research topic of the article, so the authors consider it appropriate to leave them in this form.

Comment 8: Line 113. Residueoxidation?

Response: Comment accepted. The correction has been made.   

Comment 9: Line 113. The abbreviation for liter is the capital letter L.

Response: Comment accepted. The correction has been made.   

Comment 10: Organize the materials and methods in sections. For example Oxidation process, Aging process, Mechanical characterization, etc.

Response: Comment accepted. The “Materials and Methods” section has been subdivided into “Materials”, “Oxidation process”, “Aging process” and “Mechanical characterization” subsections.

Comment 11: Revise table 1 information. There are grammatical mistakes in the title and in the content.

Response: Comment accepted. The correction has been made.

Comment 12: Figures 1-5. Include standard deviation of data.

Response: As it is known, to calculate the value of the standard deviation, one need at least 5 or 6 parallel measurements of each parameter under study. In this work, we did not set out to give a statistical assessment of the studied characteristics.

Reviewer 2 Report

The authors researched the effect of adding rubber crumbs on aging characteristics of vacuum residue. The authors tested short- and long-term aging of different samples to test their rheological properties. The results are solid and sound. The reviewer recommend for acceptance for publication before the following problems are properly addressed.

  1. The relationship between vacuum residue and asphalt is not clearly expressed in the full text. Sometimes asphalt and residue are used in the description of experimental samples.
  2. Background description uses “in the work” to make a brief list and summary of other people's research.
  3. “The complex shear modulus decreases with the temperature increase. ” The third sentence in the abstract describes the missing object.
  4. Introduction: “However, there is another modification option when the modifier is added not to the prepared bitumen, but to the bitumen raw material –vacuum residue, after which the oxidation process is carried out.” The statement expression is incomplete.
  5. Part II: materials and methods. Materials and research methods should be written separately. The instrument model is not marked in the rheological test.
  6. Table 1 and Fig 2(a), words are stuck together.
  7. The rubber chips with the concentration of 2% and 8% in Table 1 are mentioned in the article, but there is no such information in the table. Vacuum residue oxide is mentioned in the text, but rubber bitumen binder is used in the table
  8. The full text is exploring the effect of temperature on modulus,“Fatigue cracks are formed on the surface of a road asphalt concrete pavement at medium temperatures over a long service life, in connection with which the average design temperature of bitumen is determined. ” page 6. medium temperatures. What is the specific temperature range?
  9. 1 tests the shear modulus at different temperatures, and the temperature affects the shear modulus of the material. However, the coordinate temperatures of Fig. 1(a) and Fig. 1(b) are different. Is it scientific to compare the modulus data?
  10. Fig 3 and Fig 2(a),Fig 1(a): abscissa is inconsistent. There are no scale marks on the coordinate axes of Fig. 3, Fig. 4 and Fig. 5.
  11. Information in Fig. 5 is confusing?
  12. Through rheological test, it is concluded that rubber particles are rich in carbon and interact with asphalt, which doesn’t lead to the supramolecular structure of the asphalt system. Further experiments are needed to verify this supramolecular structure.
  13. Based on the experiments, the authors finally concluded that adding rubber particles can increase the deformation resistance of the material, but how is this beneficial to the use of roads is missing.

Author Response

Responses to Reviewer 2’s Comments

Comment 1.The relationship between vacuum residue and asphalt is not clearly expressed in the full text. Sometimes asphalt and residue are used in the description of experimental samples.

Response. In the research, the vacuum residue is used as a feedstock for the production of bitumen. Bitumen is a product of the vacuum residue oxidation. Corrections have been made to distinguish between the two terms.

Comment 2.Background description uses “in the work” to make a brief list and summary of other people's research.

Response. In the “Introduction” section, the word “work” is replaced by the words“research” and“paper”.

Comment 3.“The complex shear modulus decreases with the temperature increase”. The third sentence in the abstract describes the missing object.

Response.The 3rd sentence in the abstract was supplemented with the object of study “the vacuum residue and its oxidation products”.

Comment 4.Introduction: “However, there is another modification option when the modifier is added not to the prepared bitumen, but to the bitumen raw material - vacuum residue, after which the oxidation process is carried out”. The statement expression is incomplete.

Response. The authors of this proposal wanted to point out the peculiarity of this study. Unlike other researches where bitumen is modified, in this paper, the vacuum residue is first modified, then it is oxidized to obtain a bitumen.

Comment 5.Part II: materials and methods. Materials and research methods should be written separately. The instrument model is not marked in the rheological test.

Response. Materials and research methods are written separately. The dynamic shear rheometer model is marked.

Comment 6.Table 1 and Fig 2(a), words are stuck together.

Response.The words in the titles of Table 1 and Figure 2(a) are written separately.

Comment 7.The rubber chips with the concentration of 2% and 8% in Table 1 are mentioned in the article, but there is no such information in the table. Vacuum residue oxide is mentioned in the text, but rubber bitumen binder is used in the table.

Response.The text specifically mentions the product of the vacuum residue oxidation with the addition of 2 and 8% rubber crumb, since Table 1 shows the characteristics of this particular product. The product with this amount of rubber crumb was chosen as optimal in terms of penetration and softening point in order to further determine other characteristics, including rheological ones. The product of oxidation of the vacuum residue with crumb rubber is commonly called a rubber-bitumen binder.

Comment 8.The full text is exploring the effect of temperature on modulus,“Fatigue cracks are formed on the surface of a road asphalt concrete pavement at medium temperatures over a long service life, in connection with which the average design temperature of bitumen is determined”.  page 6. medium temperatures. What is the specific temperature range?

Response.As can be seen from theTable 2, as a result of short-term aging, the maximum design temperature range for the vacuum residue oxidation product with the addition of crumb rubber is from 65.1 to 68.8°C.

Comment 9.1 tests the shear modulus at different temperatures, and the temperature affects the shear modulus of the material. However, the coordinate temperatures of Fig. 1(a) and Fig. 1(b) are different. Isitscientifictocomparethemodulusdata?

Response.The standard procedures for short-term and long-term aging differ in the temperature range at which the shear modulus is determined. This is established by the standards, so the tests are carried out at such temperatures, respectively, the shear modulus values ​​are compared at a given temperature range.

Comment 10.Fig 3 and Fig 2(a), Fig 1(a): abscissa is inconsistent. There are no scale marks on the coordinate axes of Fig. 3, Fig. 4 and Fig. 5.

Response.The values of the abscissa axis in the indicated figures are given in one range and the labels of the scales of all axes are indicated.

Comment 11.Information in Fig. 5 is confusing?

Response.The reviewer’s comment is not clear. There is a term “black diagrams”, which is accepted when determining the rheological characteristics of bitumen. The description of the figure is given in the text.

Comment 12. Through rheological test, it is concluded that rubber particles are rich in carbon and interact with asphalt, which doesn’t lead to the supramolecular structure of the asphalt system. Further experiments are needed to verify this supramolecular structure.

Response. The sentence summarizes an attempt at rationalizing the observed behavior in terms of physico-chemical interactions among the chemical species involved. The fact that crumb rubber particles can be well accommodated in the organic matrix of the bitumens thanks to their organic nature is a basic chemical reasoning (apolar materials are compatible with apolar materials). The fact that the structure is reinforced is instead the clue of the whole manuscript: a higher G* implies a higher amount of mechanical energy that can be stored under mechanical stimulus. The bitumen structure, made of supramolecular structures assemblying at various length scales, is described in the cited reference (ref [21] unrevised version). The final argument on the role of amphiphilic resins is supported by the cited references(refs [22] and [23] unrevised version, but plenty of works are present in the literature). Details on the framework we are suggesting are present in the work by Tarsi et all “Study of Rubber-REOB Extender to Produce Sustainable Modified Bitumens” published in Applied Sciences Appl. Sci. 2020, 10, 1204 which showed the ability of the crumb rubber to swell itself and to adsorb the maltene part of the bitumen, additionally in the reference [Szerb et all. Highly stable surfactant-crumb rubber modified bitumen: NMR and Rheological investigation. Road Materials and Pavement Design 10.1080/14680629.2017.1289975Volume 19(5)(2018), Pages 1192-1202] it was observed that crumb rubber (CR) particles are swollen by the absorption of light fractions of bitumens, resulting in an increase in the volume of CR particles, with consequent reduction in their interparticle distance and a hindering of the dynamical processes of the bitumen molecules (structuring) and the overall increase in material viscosity and stiffness. Thanks to all these reasoning, we believe that further experimental work is unnecessary to support this rationalization, and would therefore go beyond the scope of the manuscript. However, in the light of the referee’s comment some additional comment is added in the revised version.

Comment 13.Based on the experiments, the authors finally concluded that adding rubber particles can increase the deformation resistance of the material, but how is this beneficial to the use of roads is missing.

Response.Increasing the deformation resistance of bitumen leads to an increase in compressive strength, shear resistance and shear adhesion of asphalt concrete mixtures, which improves the quality of roads.

Reviewer 3 Report

This paper refers to a well decribed experimental setup that is relevant in the current bitumen context. The scope is limited but the results seem clear.

The reading was made somewhat difficult because there were a lot of spaces missing in the text, with a lot of merged words.

Author Response

Responses to Reviewer 3’s Comments

Comment : The reading was made somewhat difficult because there were a lot of spaces missing in the text, with a lot of merged words

Response: Comment accepted. The text of the manuscript is fully verified. All merged words are separated.

Round 2

Reviewer 1 Report

The comments have been attended to. However, there are still issues that authors must consider prior to acceptance.

-The authors are not following correctly the journal instructions for authors. Revise how the research manuscript sections must be organized.

-Authors responded to the comment related to including standard deviation on data shown in figures 1-5 that they did not set out to give a statistical assessment of the studied characteristics. So, does it mean authors only carry out one experiment for each trial? How authors can assure the experiments are reproducible? 

-Include the standard deviation of data. If authors do not have such information, I consider the manuscript must be rejected since they can not prove the results are reproducible.